# Metrological Evaluation of the Demosaicking Effect on Colour Digital Image Correlation with Application in Monitoring of Paintings

**DOI:** 10.3390/s22197359

**Published:** 2022-09-28

**Authors:** Athanasia Papanikolaou, Piotr Garbat, Malgorzata Kujawinska

**Affiliations:** 1Faculty of Mechatronics, Warsaw University of Technology, 08-525 Warsaw, Poland; 2Faculty of Electronics and Information Technology, Warsaw University of Technology, 00-665 Warsaw, Poland

**Keywords:** colour digital image correlation, demosaicking, cultural heritage, natural texture, canvas paintings, DIC metrology

## Abstract

A modified 3D colour digital image correlation method (3D cDIC) is proposed for efficient displacement measurements of colour objects with natural texture. The method is using a separate analysis of correlation coefficient (sigma) value in the RGB channels of CCD cameras by utilising local information from the channel with the minimum sigma. In this way, merged U, V and W displacement maps are generated based on the local correlation quality. As the proposed method applies to colour filter array cameras, the images in RGB channels have to undergo a demosaicking procedure which directly influences the accuracy of displacement measurements. In the paper, the best performing demosaicking methods are selected. The metrological analysis of their influence on the results of canvas paintings investigations obtained by unmodified and modified 3D cDIC processing is presented.

## 1. Introduction

Three-dimensional digital image correlation (3D DIC) is a full-field optical method that enables measurements of object’s in-plane and out-of-plane displacements with high resolution and accuracy while at the same time being noncontact and noninvasive [1]. Recently, 3D DIC has gained more and more attention in the field of heritage science, as it supports the preservation and conservation of cultural heritage objects (CHOs) through quantitative monitoring of their structural health. However, a standard DIC implementation requires an object with a surface characterised by a good contrast texture, a criterion that CHOs do not always meet. Typically, DIC is performed by adding an artificial texture (i.e., graffiti spray) to the object in order to create a random speckle pattern on its surface. Then, monochrome cameras are used for the recording of the data followed by a calculation of displacements and strain maps by accurately tracking the positioning of the speckles for varying loading conditions. There are numerous successful applications of 3D DIC mainly in the field of engineering [1,2,3,4,5]. Apart from engineering applications, 3D DIC has also been reported to efficiently track displacements and strains in CHOs, such as monitoring humidity changes on canvas paintings [6,7] and parchments [8] and investigations of textiles [9] and wooden structures [10] to mention just a few.

However, the studies of CHOs using DIC are less straightforward, as their surface modifications are prohibited and the correlation has to be achieved based on the natural texture of CHOs’ surfaces. In some cases, the natural inherent texture of an object formed, i.e., by brush strokes and discolourations or intensity and colour variations, might be sufficient. While in other cases this far from the ideal speckle pattern is making the application of DIC a challenging task.

In general, when performing DIC, a monochrome camera is preferable over a colour camera of the same resolution [1,11]; however, recently, several possibilities of applying a DIC technique using colour cameras instead of monochrome ones have been discussed [11,12,13]. Such a technique is called colour digital image correlation (cDIC).

Current colour cameras work either in full-colour or interpolating mode. The former are known as true-colour or three-CCD cameras. They acquire all colour components for each pixel, which most often is realised by dividing the spectrum into three (red, green and blue (RGB)) components using a specially designed prism and three CCDs [14,15]. The acquired data can be used as captured in each full resolution R,G and B channels; however, true-colour cameras are expensive and only a few models adopt this technique. For the latter mode, known as colour filter array (CFA) or Bayer cameras [16], instead of acquiring three colour components in each location, it suffices to sample them on a regular grid and estimate the missing ones by interpolating data from neighbour pixels. Therefore, the data acquired with CFA cameras cannot be used “as is”, but a postprocessing step, known as demosaicking is required to reconstruct the three continuous RGB colour components starting from raw data. In both cases, the three RGB channels can be used separately, or they can be combined to create a monochrome image and become the input information in the conventional processing pipeline in digital image correlation [1]. However, this pipeline does not utilise efficiently the additional knowledge about an object texture hidden in the spectral information of RGB images. Therefore, in this paper we propose an important modification of the DIC data processing path, which uses three 3D DIC displacement calculation channels based on individual R,G and B images and merge the final displacement maps from the individual colour channels based on the minimum correlation coefficient values.

This new, active approach to DIC analysis can be employed for images captured by both full-colour and interpolating mode cameras; however, true-colour cameras are expensive and only a few models adopt this technique, while high-quality, high-definition and wide-dynamic-range CFA consumers cameras are easily available at low cost [11]. These facts directly translate into practical reasons for the interest in using CFA cameras for measurement applications in the field of 3D documentation, conservation and monitoring of cultural heritage objects. Therefore, in our work, we focus on CFA cameras’ implementation in the new data processing pipeline for cDIC.

As already mentioned the image captured by a CFA camera requires the implementation of a demosaicking procedure which, depending on the method, can have a significant effect on the accuracy of displacements calculated in DIC [17]. The first approach to study the influence of a variety of demosaicking algorithms on the traditional cDIC pipeline processing was presented in [13]. In this work, we propose for the first time a methodology for evaluating the performance of selected demosaicking algorithms in DIC applications in terms of both minimisation of correlation uncertainties and displacement errors. The methodology is implemented in both traditional and new cDIC data processing paths.

The analysis and experiments, in contrast to the other works devoted to cDIC (colour digital image correlation), are focused on the monitoring of objects with natural colour texture, which fulfil the requirements posed for cultural heritage objects. The best approach is selected based on the minimisation of the calculated displacements errors in order to have the maximum possible accuracy.

Finally, we aim to compare the DIC results given by both traditional and novel DIC processing pipeline and based on the selected demosaicked algorithms. It should be emphasised that the analysis and experiments, in contrast to other works devoted to cDIC, are focused on monitoring of objects with natural colour texture (namely, canvas paintings) which fulfil the requirements posed for cultural heritage objects.

The paper is organised as follows. In Section 2, we present the fundamentals of cDIC and demosaicking methods, the algorithms selected for the tests and the proposed data processing pipeline. The description of the samples (a mock-up and two paintings), experimental setup, procedure and the evaluation criteria are presented in Section 3. The results and their discussion focused on the influence of demosaicking procedures on the correlation coefficient distribution and displacement errors as well as on a comparison of the DIC processing pipelines are given in Section 4. The efficiency of the newly proposed processing pipeline is also discussed for different types of objects and introduced displacements. Finally, the conclusions and future direction of works are presented in Section 5.

## 2. Colour DIC and Data Processing

### 2.1. Colour DIC Fundamentals and Its Modification

In conventional 2D digital image correlation method, a monochrome camera is used in order to monitor displacements and strains in objects with an artificial, random black and white pattern. The displacement and strain vectors are built by tracking the position of the speckle pattern within small image areas (subset) for the different loading conditions using correlation algorithms [1]. In 3D DIC, a pair of monochrome cameras capture sequential image pairs of a sample under varying loadings. A reference pair of images is selected, and the subsequent ones are used for the correlation analysis. The correspondence of the projection of physical points in the left and right images must be established using correlation algorithms (stereo matching). Together with the calibrated parameters of the stereo rig, 3D coordinates of measurement points are recovered with the classic triangulation method. Then, combining both the triangulation and correlation algorithms enables to calculate the 3D displacement (U,V and W) and strain maps. The accuracy of the calculations depends on the quality of the correlation, which can be evaluated with a number of optimisation criteria. In this work, the correlation coefficient was determined by the zero-normalised sum of squared differences correlation criteria (ZNSSD) [1,18,19], which is insensitive to light variations, with zero values denoting perfect correlation and one total loss of it.
(1)$$Sigma_{ZNSSD}^{2}=\sum (\frac{\bar{f_{i}}}{\sqrt{\sum {f_{i}}^{2}}}-\frac{\bar{g_{i}}}{\sqrt{\sum {g_{i}}^{2}}}{)}^{2}$$
where $\bar{f_{i}}$ and $\bar{g_{i}}$ represent the intensity of pixel i(x,y) within the given subset after subtraction of the average intensity of the subset and are defined as $\bar{f_{i}}=f_{i}-\bar{f}$ and $\bar{g_{i}}=g_{i}-\bar{g}$ for the reference and loaded images, respectively. The average value of the reference and the loaded images over the corresponding subset *n* (with size of N × N pixels) is given by: $\bar{f}=\frac{1}{n}\sum_{i=1}^{n}f_{i}$ and $\bar{g}=\frac{1}{n}\sum_{i=1}^{n}g_{i}$.

In order to achieve subpixel displacement accuracy in 3D DIC, the intensity of the image needs to be estimated for subpixel positions. This is achieved using a spline interpolation method, with a higher order of splines minimising the error associated with the interpolation, which is known as the interpolation bias. The use of interpolation algorithms for the correlation matching comes at the cost of sinusoidal periodic noise, even for higher spline orders such as the eight-tap spline interpolation used in this work. As reported elsewhere regarding the interpolation bias [1,19,20,21], the minimum error is expected for integer-and-a-half-pixel positional shifts, while for the rest of the subpixel positions, its amplitude follows a sinusoidal pattern. This interpolation error may interfere with the errors introduced by the processing of the raw data recorded by a colour camera.

As described in the introduction cDIC is most often implemented with CFA cameras, due to their wide availability and low cost. The use of CFA cameras results in the capturing of a single image composed of three (for the Bayer matrix), or more, colour channels in a sparse mosaic-like arrangement. After recording an image, demosaicking (colour interpolation) methods are used for the prediction of the intensity values at each pixel position. These methods produce either three full channel resolution images (corresponding to R, G and B) or a single channel image (Figure 1). Presently, there are three approaches in the DIC analysis of images originating from CFA cameras. For a single channel output, the DIC analysis is straightforward, through a direct calculation of the correlation and displacement maps (Figure 1, grey path). In the case where the demosaicking output comprises three channels, the most common approach is to calculate the luminance from the colour information and produce a single monochrome image (Figure 1, yellow path). Then, it is possible to proceed to the DIC analysis, which utilises fully the recorded data. An alternative is to analyse each colour channel separately, with the most prominent results expected from the green channel (Figure 1, orange path), as it typically has twice the resolution of the red and blue channels, but then, only part of the recorded information is utilised. In all of the aforementioned approaches the research interest is focused on the effect of demosaicking on the correlation coefficient factor distribution and on the resultant displacement error. It has been proven through extensive studies that the demosaicking algorithms are directly influencing DIC [11,12,17,22].

However, if we consider the demosaicked images of an object with natural, colour texture, the information about the local texture in each colour channel is a bit different. This is due to different local object scattering, which varies in function of the wavelength and translates into a different speckle distribution in each channel. This specific feature can be of significant benefit for the monitoring of CH objects, and it creates the basis for the modified processing path proposed for cDIC (Figure 1 denoted with a dashed box). With the aim to utilise the information from all the colour channels, and subsequently to improve our results, we decided to investigate an alternative processing approach. After following the standard processing path for each colour channel, we utilised the local sigma values at each pixel position to construct the merged sigma map ($\sigma_{M}$) and merged $U_{M}$, $V_{M}$, $W_{M}$ displacement maps according to Equations (Equation 2) and (Equation 3).
(2)$$\sigma_{M}\left(x,y\right)=min\left(\sigma_{R\vee G\vee B}\left(x,y\right)\right)$$
for each (*x*, *y*) pixel position and the displacements are
(3)$$U_{M}\left(x,y\right)=U\left(x_{k},y_{k}\right),\;V_{M}\left(x,y\right)=V\left(x_{k},y_{k}\right),\;W_{M}\left(x,y\right)=W\left(x_{k},y_{k}\right)$$
where
$$k=\left\{\begin{array}{cc} R\; & \mathrm{if}\;\sigma_{M}\left(x,y\right)=\sigma_{R}\left(x,y\right)\\ G\; & \mathrm{if}\;\sigma_{M}\left(x,y\right)=\sigma_{G}\left(x,y\right)\\ B\; & \mathrm{if}\;\sigma_{M}\left(x,y\right)=\sigma_{B}\left(x,y\right) \end{array}\right.$$

The criterion used for the data merging was the minimisation of the correlation value at each pixel position. The merged maps with the minimum sigma value (i.e., with best quality of texture) were generated by combining the retrieved information from the channel with the optimum correlation at each analysed pixel position. This way, we were able to build a merged correlation map as well as displacement maps.

### 2.2. Demosaicking Methods

Colour image demosaicking has been an essential issue in image processing. It plays a crucial role in producing high-quality colour imagery from a single-sensor digital camera. The demosaicking processing aims to reconstruct a full-colour image from the acquired mosaicked image by estimating the values of the other two missing colour components at each pixel position. In general, a full-colour image is composed of three primary colour components at each pixel location. The colour image demosaicking has been extensively studied and there are many algorithms available to reconstruct a full resolution colour image from individual spatially separated colour channels on the CFA matrix [11,17]. Accordingly, the purpose of demosaicking is such a transformation that it allows the DIC algorithm to make the best use of the available data from the three colour channels. Image demosaicking is the interpolation process of estimating complete colour information for an image that has been captured through a colour filter array. From this sparsely sampled colour data, a full colour image is produced by interpolating the unknown values in each colour channel. Based on the existing research [13,17,23], we decided to investigate two approaches of demosaicking algorithms: a general two-dimensional interpolation and specialised algorithms developed for colour reconstruction from Bayer pattern. These approaches are represented by five groups of demosaicking methods (Table 1) which are explained below:G0 includes two commercial solutions:−The monochrome baseline method corresponds to a pixel luminance estimation based on colour information from the nearest pixels on the CFA pattern. The values for each pixel are first converted to the YCbCr colour space. The Y component of this model represents a brightness value and is equivalent to the value that would be derived from a pixel in a monochrome sensor [24] (referred to as Monochrome for the rest of the document).−At the colour baseline format, each pixel is filtered to record only one of the colours red, green, and blue. The pixel data are then processed using the demosaicking method listed in [25]. Then, a linear interpolation is used for the conversion to monochrome images (referred to as rgb2gray Basler throughout this paper to differentiate it from Monochrome).

G0 is followed by G1–G4 which represent widely used and recognised demosaicking solutions:G1—simple separate colour channel bilinear interpolation. The simplest way to restore the missing values is to interpolate each channel separately using neighbouring values. A bilinear interpolation is the most commonly used method. This method is efficient, but images would have colour artefacts at the edges [26].G2—methods using information about edges. EA (edge-aware) and VNG (variable number of gradients) reduce colour artefacts by using edge detection [27]. In this method, the gradients near the pixel of interest are computed. In the VNG method, a set of eight gradients is calculated for each pixel in a 5 × 5 neighbourhood. The edge-sensitive demosaicking methods select the interpolation direction or estimate the weights using the CFA samples, which may lead to erroneous results since the CFA image contains less information than the full-colour image.

The methods G1 and G2 were additionally extended with a dimensionality reduction operation. Two methods are used for this purpose:A principal components analysis (PCA) finds the vector in the direction where the variance is maximal;An independent component analysis (ICA) finds the vectorscorresponding to mixed signals.

Reducing the number of colour channels allows one to minimise the impact of the error resulting from an incorrect colour reconstruction, while maintaining information from all channels. Due to the fact that for complex, coloured structures, the methods from G1 and G2 may cause additional artefacts in the image, in the form of additional edges and the creation of false colour reconstructions, it was decided to also test methods commonly used in digital photography, namely:G3—directional interpolation and decision methods homogeneity. These strategies compute two or more estimation candidates for each missing component, and the decision for the best one is made a posteriori. Among the well documented representatives of G3 are adaptive homogeinity interpolation (AHD) and adaptive residual interpolation (ARI). AHD uses three different techniques to minimise the colour artefacts [28]. The filter bank technique allows the minimisation of the aliasing. The adaptive selection of the interpolation direction for misguidance colour artefacts is used. The solution for the interpolation artefacts is combining two separate interpolations (vertical and horizontal) based on a homogeneity matrix. ARI adaptively combines two residual interpolation-based algorithms and adjust the iteration number at each pixel [29]. The PPG, AAHD, DHT and DCB algorithms are undocumented, but they represented the same group [30].

Despite the high efficiency of demosaicking natural images, the methods in G0–G3 can result in the formation of artefacts which vary depending on the displacement, rotation and scale of the examined object in the image. Therefore, we decided to also investigate the selected deep learning method:G4—APN (attention pyramid network). This method is based on a single pyramid attention mechanism [31]. This method was chosen due to the use of two mechanisms that reduce the formation of artefacts. The first is nonlocal attention, with minimise the aliasing effect. The second is scale-agnostic attention that improves the quality of the detail reconstruction. The pyramid attention is implemented using convolution and deconvolution operations. The patches are extracted from the transformed feature map to a deconvolution over the matching score.

The methods from the different groups were analysed using the different processing paths presented in Figure 1. The G0 methods were treated as single channel data (directly for the monochrome and after conversion for the baseline demosaicking—named rgb2gray for the rest of the manuscript; Figure 1 grey and yellow paths), as this was the fastest way to perform DIC using a colour camera. For the G0 methods, the data processing time as well as the DIC analysis time were minimised. The other groups were analysed based on the number of output channels. Single channel data were directly passed to DIC software (Figure 1, one channel per grey path), while multichannel data were analysed separately using the modified procedure which merged displacement fields based on a minimum local correlation criterion (Figure 1, dashed box).

## 3. Materials and Methods

### 3.1. Samples

The samples used in the context of this paper were: a mock-up of a painting (*Mock-up*) and two oil paintings, named *Flower* and *Street*, fixed in a wooden frame which ensures the stability of the canvas. They were selected based on criteria connected with their colour as well as their 2D (significant difference between *Mock-up* and the paintings) and 3D textures (significant difference between the two oil paintings) connected with the oil painting technique, i.e., the way of introducing paint on the surface. *Mock-up* had the smoothest surface out of the selected samples with an average roughness of 0.003 mm, *Street* and *Flower* had an average roughness of 0.012 mm and 0.023 mm, respectively. The *Mock-up* sample was prepared on a white canvas on which a texture was created by the application of acrylic colours from Liquitex, with the aim to create a random speckle pattern. For the preparation of *Mock-up*, the colours primary blue, primary red, phthalocyanine green and mars black from Liquitex acrylics basics were used [32]. The sample included five areas with artificial texture of different colours (Figure 2). At the top of the sample separate red, green and blue patterns were applied; the middle area consisted of a typical black pattern on a white background and at the bottom of the sample another area was covered with randomly distributed mixed red, green and blue speckles. The black and white area was expected to have a similar response for the different channels, while the rest was strongly depending on the pixel filter. *Mock-up* (Figure 2a) was used as a reference, with a random artificial texture, to investigate the performance of the demosaicking methods and their influence on the calculated displacement maps. For the validation of the results two oil paintings on canvas without the addition of any artificial texture (i.e., with their natural texture) were used. The painting presented in Figure 2b is referred to as *Flower* (with the evident 3D paint texture from the artist’s brushstrokes) and the painting in Figure 2c as *Street* (with a less pronounced 3D texture).

### 3.2. Three-Dimensional DIC System and Experimental Details

The system used for the 3D DIC measurements consisted of two colour cameras attached rigidly to a steel profile so that their optical axis crossed at an angle of 35.4 degrees. The cameras were from Basler (model: aca4112-8gc) with a resolution of 4096 × 3000 pixels and the samples were illuminated using two white light LEDs. A single pixel at the image plane of all objects had a dimension of 0.093 mm, so the expected displacement accuracy was approximately 0.005 mm. Using 3D DIC, the shape, as well as in-plane and out-of-plane displacement maps of the samples exposed to controlled shifts in the x- and z-axes were measured. The samples were positioned on a x–z micrometric stage, so that a controlled movement in both x- and z-axes was possible. The reference images were captured at the initial sample position. Then, a series of images were captured following mechanical translations which corresponded to an x direction of 1 cm with a step of 1 mm. After 10 steps of x-translation, the samples were additionally subjected to five shifts in the z direction, again with a step of 1 mm. The same procedure was repeated for all three samples. Data were recorded using the Basler RG8 pixel format, both in “tiff” and “raw” format, as well as in the monochrome mode. The DIC processing was performed for the subset of 105 pixels with a step of 7 pixels using VIC 3D-7 software. The significant size of the subset was required by the objects with natural textures. For consistency purposes, the subset and step were not optimised for each individual object but kept constant in all cases.

### 3.3. Evaluation Criteria

The most important metric for DIC measurements is the distribution of the correlation coefficient (see Equation (Equation 1)). The correlation coefficient (or sigma) denotes the accuracy of the correlation achieved at the local level across the area of interest (AoI). In the case when the local texture is of low quality (i.e., lack of random texture), sigma is expected to have high local values, and in the extreme cases where certain thresholds are exceeded, the areas are locally masked and exempted from the analysis, while globally the mean value of sigma is increased. In this work, the selected thresholding values were: 0.02 pixels for the consistency threshold, 0.05 pixels for the confidence margin and a 0.1 maximum pixel margin.

The analysis of the performance of the 3D cDIC measurements was performed within two stages. At first, we investigated the evaluation criteria based on sigma, namely: (i) the local distribution of sigma values within the selected AoI maps and (ii) the mean sigma value (<sigma>) and its standard deviation (<std>) within the AoI. Local variations of sigma influenced the local accuracy of the calculated displacements, and therefore, during the second stage, we referred to the metrics connected with the calculated displacements and their deviations from the expected values. In order to effectively detect these errors, a plane fit to the resultant displacement maps was performed, and both the intersect and slope related to the induced displacements and the experimental error, respectively, were subtracted. Thus, the residuals which were associated with the combined effect of demosaicking and interpolation bias errors could be further investigated, through a calculation of its mean value and standard deviation (<Res. Disp.>). The adjacent R-squared value was $R^{2}>0.99$ in all cases, denoting the good quality of the fitting. A further metric for the evaluation criteria was the variations in the RMSE (root-mean-square error) value of the residual displacement map (<RMSE Res. Disp.>). It allowed us to evaluate the combined demosaicking and interpolation bias error. Finally, the peak-to-valley amplitude of the modulation (P/V Amplitude Res. Disp.) in the residual maps was also calculated, as it could indicate the local interactions between the demosaicking and interpolation errors. The DIC analysis parameters and AoI were kept constant throughout all of the analysed data sets, making thus the interpolation error comparable for the different translation values, and allowing us to attribute variances to the demosaicking error.

## 4. Results

### 4.1. Correlation Coefficient Analysis

The first step towards the evaluation of the demosaicking methods was to investigate the correlation coefficient (sigma). Sigma can be used to determine the efficiency of the correlation which is linked with the accuracy of the calculated displacements. For this reason, in Figure 3, we present the comparison of the mean sigma and its standard deviation for the different demosaicking methods investigated for two known translations of an object. The first one corresponded to a translation in the x direction X5=(5±0.1) mm, while the second one to a combined shift in the x and z directions XZ5=X5+Z5, where Z5=(5±0.1 mm). These shifts were selected as they represented the middle and end of the experimental data recorded and they could reveal the propagation of the combined error of demosaicking and correlation. High sigma values were most often linked with high standard deviation values, therefore demosaicking methods for which this behaviour was detected were eliminated from further investigation.

Based on the results presented in Figure 3, the methods with a high sigma, and therefore the highest measurement uncertainty and potential loss of correlation, could be easily detected. Furthermore, we can observe that the sigma values for shifts X5 and XZ5 had comparable values. The lowest sigma values were obtained for the deep learning APN method, followed by our modification based on the channel merging process. The worst results were observed for the merged AHD, VNG PCA and VNG ICA methods. The different camera recording formats, Monochrome and Basler, without any additional interpolation were also showing acceptable sigma values, but higher than for the APN method. Results comparable to the G0 methods were also recorded for the ARI method, but again APN had a lower average sigma. The exemplary maps of sigma distribution at the *Mock-up* sample are presented in Figure 4. In the methods with the highest standard deviation, the correlation was locally lost, as is evident in Figure 4d for the worst case of the merged AHD method. The merged APN format presented lower local and global correlation values in comparison with the Monochrome and Basler methods. We can also detect that the area with the black speckle pattern showed a good correlation for both the merged APN (Figure 4a) and the Basler methods (Figure 4c), while there were no significant differences for the results obtained for the colour textures. Based on these observations, we selected to proceed further with the analysis of the merged APN and the commercially available methods, Monochrome and Basler. These cases represented both a potential improvement and the option of performing cDIC fast and with a low computational complexity.

The *Mock-up* sample had a random pattern which enabled an easy DIC analysis, but this is not always valid for cultural heritage objects, where an artificial pattern cannot be applied. Therefore, we extended our study to two oil painting samples, which had a prominent colour texture but far from ideal. At this stage, we compared all the demosaicking methods for both of the paintings, in order to reveal the ones which should be further considered for our metrological evaluation. As shown in (Figure 5), for both paintings, we observe a significant increase of mean sigma and standard deviation values. This increase is obvious due to the local variation of the texture quality, which is also indicated in the sigma distribution maps (Figure 6 and Figure 7). The optimum results (with lowest sigma) for *Street* were achieved with the merged APN method. For *Flower*, the Basler method after conversion to monochrome and the monochrome methods had the lowest sigma, with the merged APN method presenting a comparable result.

Regarding *Flower*, all methods had a similar distribution of sigma values and they exhibited lower sigma values at the centre of the painting, where the natural texture was of best quality ( Figure 6). The sigma maps of *Street* presented much bigger differences between the methods, with significantly better results detected for the merged APN method.

In order to evaluate the efficiency of the proposed pipeline, the sigma maps corresponding to the Basler after the merging process are shown in Figure 8. Comparing the sigma distribution for the merged Basler with the rgb2gray Basler method (shown in Figure 4, Figure 6 and Figure 7), we can conclude that its local values’ distribution improved for both *Mock-up* and *Street* when compared with the rgb2gray Basler and Monochrome methods. The situation was not so evident for *Flower*, which had an optimum sigma distribution for the rgb2gray Basler method, meaning the classical pipeline. This phenomenon can be linked to the enhanced 3D texture of this painting.

The main conclusions which can be drawn at this stage are:The suggested pipeline resulted into an improved sigma distribution for the samples with less 3D local texture (*Mock-up* and *Street*), while it was not so evident for the sample with a high roughness (*Flower*).The methods with aggregation colour information (G0–G2) linear such as Monochrome or with dimension reduction such as EA PCA/ICA and VNG PCA/ICA allowed us to use the data from all channels simultaneously. Unfortunately, the methods based on dimension reduction depended on the context of the image.The commonly used demosaicking methods based on a homogeneity matrix (G3) were generally better than simple interpolation methods (G1), but they were also sensitive to the local (5 × 5 neighbourhood) configuration of elementary colour pixels (*Flower* and *Street*). The decrease of the interpolation quality could be caused by a mismatch in the structure of the homogeneity matrix. It should be noted that depending on the methods from this group, different colour components were preferred.The deep learning method (G4) showed the most significant potential compared to aggregating colour information and methods testing local consistency. The two main reasons were using data from a multiresolution representation and a nonlocal approach to the homogeneity determination. We can assume that it was not an effect of overfitting to the test images because the method was trained on data unrelated to the field of activity (DIV2K [33,34] and our own data set of 300 real and synthetic images).

### 4.2. Evaluation of Demosaicking Influence on Displacement Maps

The goal of a DIC method is to deliver accurate displacement maps. The demosaicking methods influence directly the distribution of the intensity in images, so they introduce errors in the resultant displacement maps. In order to evaluate the influence of the selected demosaicking methods, we proceeded to the evaluation of the effect of demosaicking methods to the calculated displacements. These errors were hidden within the displacement maps calculated, so in order to evaluate them, we proceeded to the calculation of the residual error of selected displacements, bearing in mind that the introduced displacements had step-like constant values with a probability of small linear term. Therefore, the metrological evaluation of the applied methods was performed by computing the residuals of displacements calculated from data delivered by the selected demosaicking method after performing a subtraction of the constant (mean displacement) and linear (plane fitting) terms. In this section, we compare only the methods identified as optimal namely: merged APN, Monochrome, Basler after conversion to monochrome. Starting with *Mock-up*, we present the residual error of displacement for a selected step, the full information about the fitting quality, RMSE and the corresponding sigma value in Table 2. The example residual error maps for displacement X5 are presented in Figure 9. As mentioned before, we observed here the total error originating from both interpolation bias and demosaicking. It is clearly seen that the amplitude of the sinusoidal error generated by the interpolation bias was significantly lower for merged APN and highest for Monochrome.

We also noticed the smallest error (for all methods) in the area of the “black dot” texture, which was also in agreement with the smallest values of sigma in this region (Figure 4). To confirm the superiority of the merged APN method, we present in Table 2 the mean sigma, along with its standard deviation and the metrics for the accuracy of the displacement maps:RMSE and mean of residual displacement mapsThe P/V of the modulation in the residual displacement maps

The values of the aforementioned quantities are presented for three selected displacements, namely X5, X10 and XZ5. The merged APN method is characterised by the smallest sigma values and P/V of the modulation in the residual displacement maps, which is also clearly seen in Figure 9. It means that this method decreased significantly the interpolation bias error. We can conclude that the local accuracy of displacement increased two to three times. On the other hand, the global evaluation parameters such as RMSE of residual displacements and mean residual displacements for each translation case were comparable between the three methods considered. It was also clear when changing the values and direction of the displacement that the merged APN method provided the most accurate results, although the global RMSE parameters were comparable with Monochrome and Basler. The interpolation bias error can be seen in the form of the repeating quasi-sinusoidal modulations in the residual displacement maps. For each algorithm, this pattern changed in magnitude and frequency, depending on the quality of the texture pattern, along with demosaicking artefacts and interpolation bias errors. It is also apparent that the demosaicking method had a large effect on the amplitude of the interpolation bias error observed, with the merged APN method clearly producing less prominent interpolation bias errors in DIC than the other methods considered. The merged-APN-based algorithm showed a reduction in integer and bi-integer aliasing in comparison to the other methods tested here (Figure 9). This behaviour was repeated for different linear shifts introduced to the sample.

At the next step, we investigated how displacement are influenced by demosaicking in samples with a natural texture pattern. For this reason, we applied the same testing procedure to the *Flower* and *Street* paintings and computed the residual displacement error maps (Figure 10 and Figure 11).

In both cases, the merged APN returned more homogeneous and lower values of the residual displacement maps. It was also confirmed by the P/V amplitude of the modulations in the residual displacement, which was smaller for all three cases of paintings’ shifts (Table 2, Table 3 and Table 4). The distribution of the residual error varied between the three samples investigated herein, and this could be attributed to the difference in the individual patterns. *Mock-up* had the most prominent texture, and thus the lower errors for all methods, while *Flower* and *Street* had a natural texture and thus errors were also higher. *Mock-up* and *Flower* presented a more homogeneous residual error distribution, while *Street* had the highest modulations in the residual displacement maps. Regarding the global metrics, the mean sigma value was lower in the case of the monochrome Basler method, but the P/V amplitude of the residual displacements, along with the RMSE, was lower for the merged APN method for *Flower*. For *Street*, the sigma, RMSE and residual P/V amplitude were again lower for the merged APN method in comparison to the other two.

## 5. Discussion

In this paper, we investigated the potential application of colour DIC on samples with natural texture for the monitoring of surface displacements. We performed an in-depth analysis of the influence of colour image preprocessing on the displacement accuracy delivered by 3D DIC. We introduced and evaluated a novel data processing pipeline for cDIC, which could improve the accuracy of displacements calculated based on the natural texture of colour paintings. Furthermore, to our best knowledge, this was the first time that the influence of demosaicking procedures (including the deep-learning-based demosaicking method) was studied in reference to the resultant accuracy of displacements for samples with natural textures. The investigations were performed for five groups of demosaicking methods: baseline Basler, pixel luminance estimation based on colour information, simple separate colour channel bilinear interpolation, directional interpolation and decision methods homogeneity and a deep learning method. The methods with a single channel output were analysed using the typical cDIC processing path. The methods with three channel outputs were analysed using the alternative pipeline proposed by us, which included merging the information from the individual colour channels using the minimum of the local correlation coefficient value as the criterion. The effect of the demosaicking on the displacements was evaluated based on local and global values of sigma, along with a variety of metrics of residual displacement maps. We proved that the local accuracy of the displacement calculation was improved at least two times, using the merged APN method, when compared with typical colour image preprocessing methods applied for cDIC.

This work can serve as a reference for the audience that wishes to implement 3D cDIC using widely available colour cameras. From our research, two possible processing paths arose. The first consists of using directly colour images captured for the calculation of 3D DIC results. This option is straightforward, requires less processing time and can be achieved by inexperienced users, but generates displacement results with higher errors. We showed that the colour image preprocessing offered by camera produces was well matched to DIC requirements for images with a reasonable texture. It was proven that it could be applied both in samples with an artificial colour random speckle pattern as well as with a natural texture but at the expense of displacement accuracy.

The second option proposed in this paper was to apply the APN demosaicking method and channel merging, in order to calculate a result with minimised displacement errors. This option provides a result with minimised displacement errors when compared with conventional cDIC preprocessing, but at the expense of a more complex processing and longer computational time. The modified 3D cDIC procedure is well matched to the requirements connected with a variety of cultural heritage objects investigations. It can offer better coverage of the measurements over surfaces with a variable quality of natural texture, while maintaining an acceptable accuracy of the results. Therefore, it should be preferred in cases where a high resolution and accuracy in the calculated quantities is required.

Colour DIC may significantly help the analysis of objects with a natural colour texture. Therefore, in future works we are aiming to extend our study in other materials and use cDIC for the in situ monitoring of cultural heritage objects such as paintings, parchments and textiles. It is also possible to expand the results towards a wider spectral range (UV-IR) and generate multispectral 3D DIC results by using an appropriate illumination source and detector. In this case, it would be expected to have unique texture characteristics at each spectral band, depending on the wavelength, making it possible to apply a methodology similar to ours for more than three spectral bands.

## Figures and Tables

**Figure 1 sensors-22-07359-f001:**
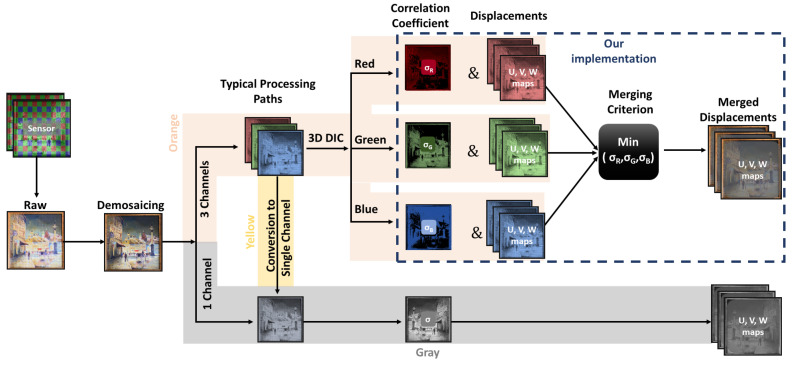
Schematic representation of data processing path.

**Figure 2 sensors-22-07359-f002:**
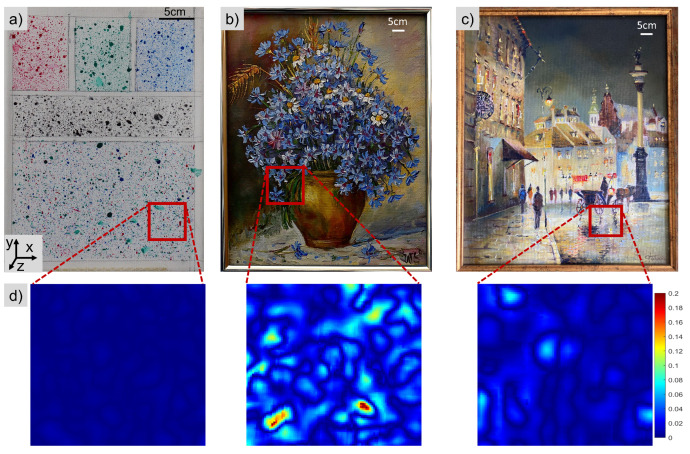
Samples (**a**) *Mock-up* with random artificial texture, and the oil paintings (**b**) *Flower* and (**c**) *Street*; (**d**) the local maps of roughness in the example areas denoted at the samples with red rectangles.

**Figure 3 sensors-22-07359-f003:**
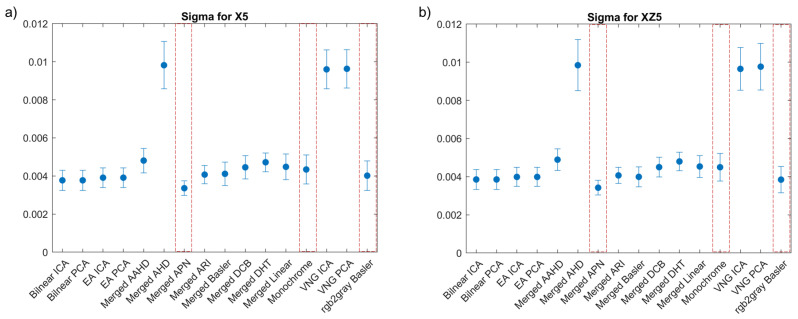
Mean correlation coefficient (sigma) and standard deviation values as calculated for (**a**) X5 shift and (**b**) XZ5 shift. Red boxes indicate the selected methods.

**Figure 4 sensors-22-07359-f004:**
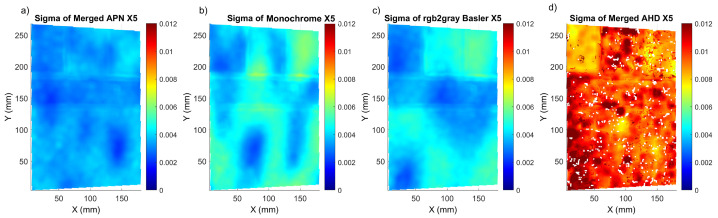
Correlation coefficient (sigma) maps corresponding to X5 shift for: (**a**) merged APN, (**b**) Monochrome, (**c**) rgb2gray Basler and (**d**) merged AHD methods.

**Figure 5 sensors-22-07359-f005:**
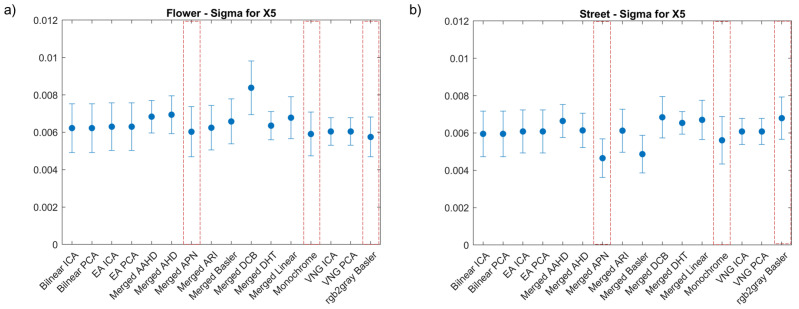
Correlation coefficient (sigma) maps corresponding to X5 shift for: (**a**) *Flower* and (**b**) *Street* paintings. Red boxes indicate the selected methods.

**Figure 6 sensors-22-07359-f006:**
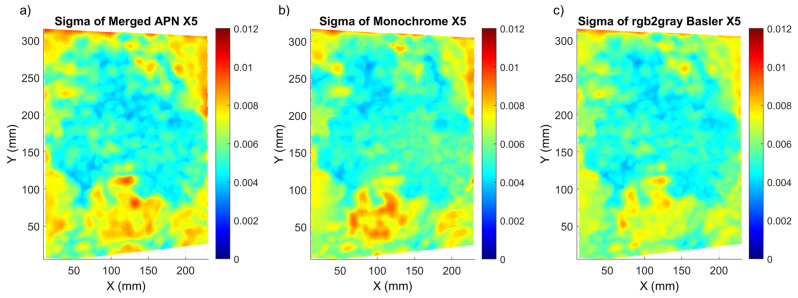
Sigma distribution for the *Flower* painting corresponding to the X5 shift for the (**a**) merged APN, (**b**) Monochrome and (**c**) rgb2gray Basler methods.

**Figure 7 sensors-22-07359-f007:**
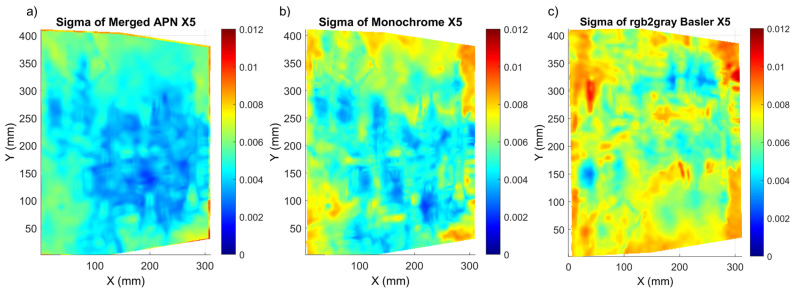
Sigma distribution for the *Street* painting corresponding to shift X5 for the (**a**) merged APN, (**b**) Monochrome and (**c**) rgb2gray Basler methods.

**Figure 8 sensors-22-07359-f008:**
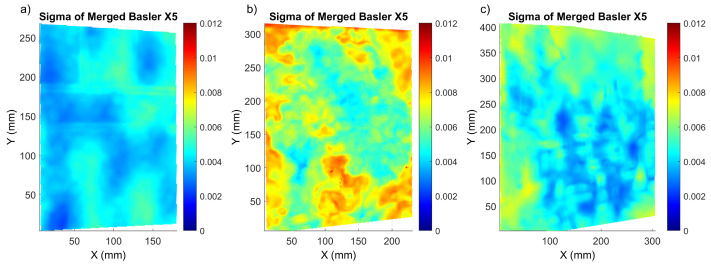
Sigma distribution for (**a**) *Mock-up*, (**b**) *Flower* and (**c**) *Street* corresponding to shift X5 after the merged Basler method.

**Figure 9 sensors-22-07359-f009:**
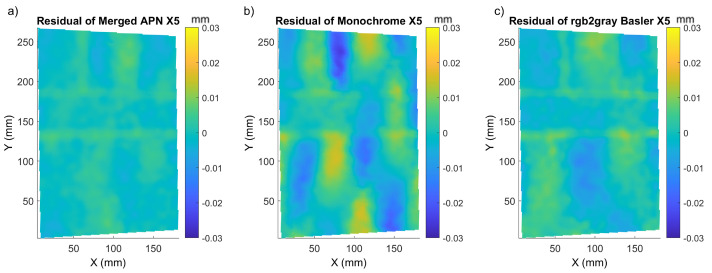
X5 residual displacement maps of *Mock-up*, for: (**a**) merged APN, (**b**) Monochrome and (**c**) Basler converted to monochrome.

**Figure 10 sensors-22-07359-f010:**
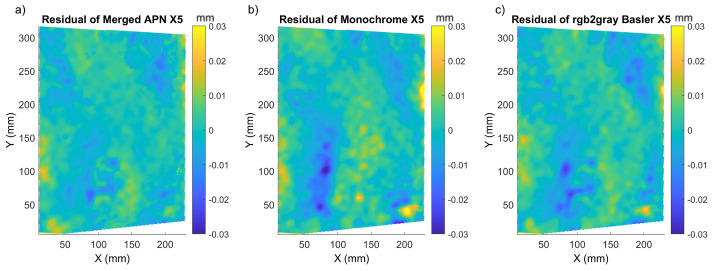
X5 residual displacement maps for *Flower* corresponding to: (**a**) merged APN, (**b**) Monochrome and (**c**) Basler converted to monochrome.

**Figure 11 sensors-22-07359-f011:**
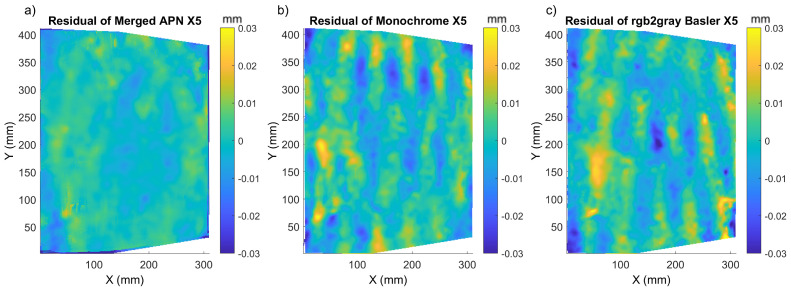
X5 residual displacement maps for *Street* corresponding to: (**a**) merged APN, (**b**) Monochrome and (**c**) Basler converted to monochrome.

**Table 1 sensors-22-07359-t001:** Demosaicking methods.

Group No.	Method	Interpolation Type	# Output Channels	Computational Complexity
0	Baseline monochrome—Basler	Nearest neighbour	1	Small
	Baseline demosaicking—Basler	Multi. interpolation + enhancement	3	Small
1	Bilinear ICA	Bilinear	1	Mid
	Bilinear PCA	Bilinear	1	Mid
2	EA ICA	Edge adaptive bilinear	1	Mid
	EA PCA	Edge adaptive bilinear	1	Mid
	VNG ICA	Gradients + bilinear	1	Mid
	VNG PCA	Gradients + bilinear	1	Mid
3	PPG	Adaptive	3	High
	AHD	Adaptive	3	High
	AAHD	Adaptive	3	High
	DHT	Adaptive	3	High
	ARI	Adaptive	3	High
4	APN	Deep learning	3	High

**Table 2 sensors-22-07359-t002:** Demosaicking evaluation for *Mock-up*.

Method and Shift	<Sigma> (NN)	RMSE Res. Disp. (mm)	<Res. Disp.> (mm)	P/V Amplitude Res. Disp. (mm)
Merged APN X5	0.0034±0.0004	0.0203	0.00005±0.00246	0.009
Monochrome X5	0.0043±0.0008	0.0215	−0.00020±0.00725	0.031
rgb2gray Basler X5	0.0040±0.0008	0.0164	−0.00002±0.00221	0.02
Merged APN X10	0.0033±0.0004	0.0311	0.00003±0.00226	0.007
Mononochrome X10	0.0044±0.0008	0.0319	0.00001±0.00713	0.033
rgb2gray Basler X10	0.0038±0.0007	0.0304	0.00004±0.00199	0.018
Merged APN XZ5	0.0034±0.0004	0.0322	−0.00006±0.00182	0.009
Mononochrome XZ5	0.0045±0.0007	0.0329	−0.00006±0.00666	0.034
rgb2gray Basler XZ5	0.0039±0.0007	0.0320	−0.00001±0.00379	0.019

**Table 3 sensors-22-07359-t003:** Demosaicking evaluation for *Flower*.

Method and Shift	<Sigma> (NN)	RMSE Res. Disp. (mm)	<Res. Disp.> (mm)	P/V Amplitude Res. Disp. (mm)
Merged APN X5	0.0060±0.0013	0.0117	0.00061±0.02405	0.019
Monochrome X5	0.0059±0.0012	0.0124	0.00048±0.02461	0.039
rgb2gray Basler X5	0.0057±0.0011	0.0121	0.00041±0.02430	0.035
Merged APN X10	0.0062±0.0013	0.0294	0.00477±0.20020	0.026
Monochrome X10	0.0064±0.0012	0.0299	0.00303±0.14069	0 .034
rgb2gray Basler X10	0.0060±0.0012	0.0296	0.00307±0.14033	0.031
Merged APN XZ5	0.0062±0.0013	0.0287	0.00331±0.14141	0.027
Monochrome XZ5	0.0065±0.0011	0.0291	0.00329±0.14147	0.046
rgb2gray Basler XZ5	0.0060±0.0011	0.0290	0.00317±0.14151	0.039

**Table 4 sensors-22-07359-t004:** Demosaicking evaluation for *Street*.

Method and Shift	<Sigma> (NN)	RMSE Res. Disp. (mm)	<Res. Disp.> (mm)	P/V Amplitude Res. Disp. (mm)
Merged APN X5	0.0047±0.0010	0.0208	−0.00014±0.00438	0.017
Mononochrome X5	0.0056±0.0013	0.0381	−0.00012±0.00773	0.039
rgb2gray RGB X5	0.0068±0.0011	0.0385	−0.00010±0.00820	0.049
Merged APN X10	0.0047±0.0010	0.0655	0.00051±0.02149	0.017
Mononochrome X10	0.0057±0.0012	0.0759	−0.00042±0.00893	0.041
rgb2gray RGB X10	0.0069±0.0011	0.075	−0.00062±0.00964	0.057
Merged APN XZ5	0.0047±0.0011	0.0541	0.00070±0.03299	0.027
Mononochrome XZ5	0.0058±0.0012	0.0617	−0.00043±0.01043	0.034
rgb2gray RGB XZ5	0.0069±0.0011	0.0634	−0.00071±0.01059	0.065

## Data Availability

The data used for this study can be accessed upon request.

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
