# Peer review of "Metrological Evaluation of the Demosaicking Effect on Colour Digital Image Correlation with Application in Monitoring of Paintings"

_sensors, 2022, doi:10.3390/s22197359_

Round 1
Reviewer 1 Report
It is necessary to highlight the experimental results and the main advantages of the proposed method.
It is necessary to indicate in the conclusion of the article the main results obtained and the advantages of the proposed method.
Reviewer 2 Report
This manuscript investigates the effects of demosaicing on color digital image correlation (cDIC). The demosaicing algorithm is classified according to its own criteria such as complexity and algorithm characteristics, and performance evaluation is performed in terms of cDIC. In addition, a modified 3D cDIC method is proposed and demosaicing with the best performance is selected.
However, many weaknesses exist in the current manuscript.
1) The author needs to compare with signal processing-based state-of-the-art methods such as residual interpolation, which is more convincing for method evaluation.
2) It is also necessary to study and compare the demosaic approach based on deep learning other than APN.
3) The comparisons are only conducted on three samples. The author should conduct more experiments to demonstrate the effectiveness.
4) Some references to demosaicing algorithms such as AHD are missing.
5) There are some typos, grammar errors in the paper.
6) Above all, the contribution of the manuscript is limited and not highlighted.
Reviewer 3 Report
The paper “Metrological evaluation of the demosaicing effect on color Digital Image Correlation with application in monitoring of paintings” presents a study on the influence of various existing demosaicing methods on displacement accuracy of 3D DIC. In addition, the authors propose also an alternative processing path for three channel images consisting in merging individual color channels using minimum of the local correlation coefficient.
The paper is well written, and the methodology is clearly presented. The authors use three different images to conduct the investigation, one containing artificial random color texture and two oil paintings. The results are discussed against these three samples, motivated by the targeted applications in the field of conservation of cultural heritage objects as paintings or textiles. The study is clearly organized, and the results are deeply analyzed. However, there are some parts that should be improved as following.
1. The purpose and the outcome of the methodology should be better motivated in the introductory section.
2. In the related work presentation, the authors must clarify the differences between the presented work and the reference [13] published by them in 2021, which also aims “This work's main objective is to select the best demosaicing algorithms for the spectral and monochromatic channels to minimize displacement and strain reconstruction errors (STD and P/V) [13].”.
3. The experiments must be better organized and explained. In the Section 3.1. it is not clear why the authors choses the two paintings – Flowers and Street – to test the method. Why these two? What strengths/weaknesses of the method they expect to emphases considering each of them, individually? They seem to be randomly selected as “natural textures!?”.
4. The discussions section must contain some practical conclusions of the presented study. As the authors declared as one important objective the study/selection of demosaicing methods with the best results on displacement accuracy, they must consider the type of objects/material where each method provides best results. Observations as [403] “Mockup and Flower present a more homogeneous residual error distribution, while Street painting has the highest modulations in the residual displacement maps. Regarding the global metrics, the mean sigma value is lower in the case of the monochrome Basler method, but the P/V amplitude of the residual displacements along with the RMSE is lower for the merged APN method for Flower. For Street, sigma, RMSE and residual P/V amplitude are again lower for the merged APN method in comparison to the other two.” suggests that various methods are suitable in different practical use cases. Identification of such cases is more interesting than a general conclusion as [432] “This option is straightforward, requires less processing time and can be achieved by non experienced users.”
Round 2
Reviewer 2 Report
The GBTF or RI algorithm is known to have good overall performance, but it is difficult to understand that artifacts appear in the experimental image. It is recommended to apply the ARI (Adaptive Residual Interpolation) algorithm, which is open source and is known to have the best performance.
Reviewer 3 Report
The revised version of the paper “Metrological evaluation of the demosaicing effect on colour Digital Image Correlation with application in monitoring of paintings” clarifies better some aspects of the proposed method and the practical purpose of the algorithm. Most of my comments and suggestion where successfully addressed by the authors. Fig 2 was improved to better explain the outcome of the method relatively to the average roughness of the material of the targeted sample. The conclusions now clarify better the practical use of the presented method. The paper still contains some ambiguity regarding the substance of the differences between the presented paper and reference [13], but the overall improved quality of the manuscript allows me to recommend it for publication.
